# Characteristics and clinical outcomes of patients with kidney failure of unknown aetiology from ANZDATA registry

Lucy S. Wang[1], Venkat Vangaveti[2,3], Monica S. Y. Ng[1,4,5‡] *, Andrew J. Mallett [2,3,4,6‡] *

**1** Kidney Health Service, Royal Brisbane and Women's Hospital, Brisbane, Queensland, Australia, **2** College of Medicine and Dentistry, James Cook University, Townsville, Queensland, Australia, **3** Townsville Institute of Health Research and Innovation, Townsville University Hospital, Douglas, Queensland, Australia, **4** Faculty of Medicine, University of Queensland, Brisbane, Queensland, Australia, **5** Conjoint Internal Medicine Laboratory, Chemical Pathology, Pathology Queensland, Brisbane, Queensland, Australia, **6** Institute for Molecular Biosciences, University of Queensland, Brisbane, Queensland, Australia

‡ MSYN and AJM are contributed equally to this work as senior authors
* monica.ng@health.qld.gov.au (MSYN); andrew.mallett@health.qld.gov.au (AJM)

**Data Availability Statement:** Data Availability Statement The authors confirm that all data underlying the findings are fully available upon request and without restriction. The primary

## Abstract

### Introduction

Kidney failure of unknown aetiology (uESKD) is also heavily location dependent varying between 27% in Egypt to 54% in Aguacalientes, Mexico. There is limited information about the characteristics of people with uESKD in Australia and New Zealand, as well as their clinical outcomes on kidney replacement therapy.

### Methods

Data on people commencing kidney replacement therapy 1989–2021 were received from the Australia and New Zealand Dialysis and Transplant (ANZDATA) registry. Primary exposure was cause of kidney failure–uESKD or non-uESKD (known-ESKD). Primary outcome was mortality. Secondary outcome was kidney transplantation. Dialysis and transplant cohorts were analysed separately. Cox Proportional Hazards Regression models were used to evaluate correlations between cause of kidney failure and mortality risk. Subgroup analyses were completed to compare mortality risk in people with uESKD to those with diabetic nephropathy, autosomal dominant polycystic kidney disease (ADPKD), glomerular disease and other kidney diseases.

### Results

This study included 60,448 people on dialysis and 20,859 transplant recipients. 1-year, 3-year and 5-year mortality rates in people with uESKD on dialysis were 31.6%, 58.7% and 77.2%, respectively. 1-year, 3-year and 5-year mortality rates in transplant recipients with uESKD were 2.8%, 13.8% and 24.0%, respectively. People with uESKD on dialysis had a higher mortality risk compared to those without uESKD on univariable and multivariable analyses (adjusted hazard ratio [AHR] 1.10, 95% CI 1.06–1.16, p<0.001). Transplant recipients with uESKD have a higher mortality risk compared to those without uESKD on

dataset for this manuscript was generated and made available to the authors by the Australia and New Zealand Dialysis and Transplant (ANZDATA) Registry, Adelaide, Australia. Data used in this study belongs to the ANZDATA registry. Data stored in ANZDATA is collected and stored on behalf of all Australian and New Zealand renal units. The ANZDATA data usage agreement between the ANZDATA Registry and the authors does not allow the authors to make the data publicly available. The authors confirm that all data underlying the findings can be obtained without restriction directly from the ANZDATA Registry on request (email address requests@anzdata.org.au, website https://www.anzdata.org.au/anzdata/services/data-policies/). The authors of this paper did not access the data via special access privileges and only gained access to the data after the data request was approved by the ANZDATA.

**Funding:** The author(s) received no specific funding for this work.

**Competing interests:** M.S.Y.N. has received research grants and travel sponsorships from Avant Foundation and postdoctoral research fellowship from Royal Brisbane and Women's Hospital Foundation. This does not alter our adherence to PLOS ONE policies on sharing data and materials.

univariable and multivariable analyses (AHR 1.17, 95% CI 1.01–1.35, p<0.05). People with uESKD had similar likelihood of kidney transplantation compared to people with known-ESKD.

## Conclusion

People with uESKD on kidney replacement therapy have higher mortality risk compared to people with other kidney diseases. Further studies are required to identify contributing factors to these findings.

## Introduction

Kidney failure of unknown aetiology (uESKD) is also heavily location dependent varying between 27% in Egypt to 54% in Aguacalientes, Mexico [1–3]. uESKD is defined as kidney failure cases where there other causes of kidney diseases such as diabetes, hypertension, glomerular disease have been excluded as potential causes [1]. Inroads to identify causes of uESKD have been made with advances in genetic kidney diagnoses, however, 80% of initially uESKD remains without a causal diagnosis [4]. Therefore, there are no cause-specific treatment options, disease recurrence risks are unquantifiable and referral for transplant may be delayed due to uncertain recurrence risk. There is limited information about the characteristics of this group of patients; as well as their clinical outcomes after KRT initiation, including mortality risk and likelihood of kidney transplantation. Information about clinical outcomes of people with uESKD is essential to guide disease prognostication, patient counselling and KRT modality selection. This Australian and New Zealand Dialysis and Transplant (ANZDATA) registry analysis aimed to profile, at a population level, the characteristics, mortality risk and likelihood of kidney transplantation for people receiving KRT due to uESKD. We hypothesised that people with uESKD would have similar mortality risk and reduced likelihood of kidney transplantation compared to people with known-ESKD.

## Materials and methods

### Study population

This population-based cohort study included people over 18 years old who initiated kidney replacement therapy in Australia and New Zealand between 1 January 1989–31 December 2021. Demographic, comorbidity, kidney failure and outcome data were extracted from the Australia and New Zealand Dialysis and Transplant (ANZDATA) registry in de-identified format and accessed on 1st April 2022. This access did not include access to information that could identify individual participants during or after data collection. The dialysis cohort included all adults who received dialysis as sole kidney replacement therapy modality (Fig 1). The transplant cohort included all adults who received a kidney transplant (Fig 1). Ethics approvals were received from ANZDATA executive (Request ID: 42579) and Metro North Human Research and Ethics Committee (Reference: LNR/2019/QRBW/58238). Written informed consent to the ANZDATA Registry was not required as a national quality assurance registry program. This study was reported per the Strengthening the Reporting of Observational Studies in Epidemiology (STROBE) guidelines [5].

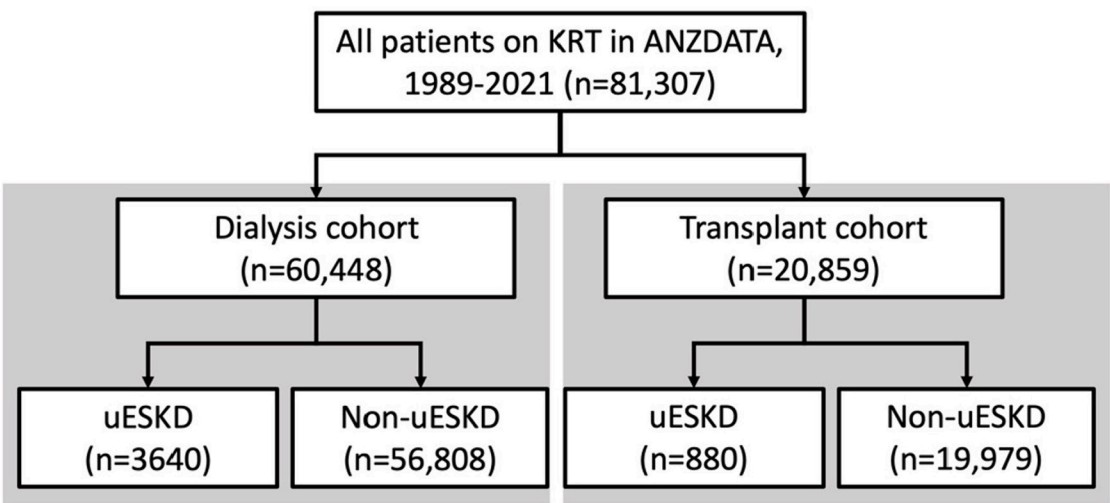

**Fig 1. Flow chart demonstrating stratification of patient cohorts.**

## Variables

Primary exposure was kidney disease type classified as *kidney failure of unknown aetiology* (uESKD) or *non-uESKD* (known-ESKD) based on kidney failure cause codes in ANZDATA. Cases not caused by diabetic nephropathy, glomerular disease, hypertension or any other identifiable cause are classified as uESKD in ANZDATA. In ANZDATA, cause of kidney failure is denoted by treating kidney specialist based on clinical features and may not be biopsy- nor genetically- proven. Primary outcomes measures were mortality in the dialysis and transplant cohort; and kidney transplantation in all patients on KRT. Age and comorbidities were recorded at time of dialysis commencement for dialysis cohort and time of kidney transplant in transplant cohort. Age was classified by 10 year intervals. Comorbidities included diabetes, coronary artery disease and peripheral vascular disease which were denoted by treating kidney specialist. First KRT modality included haemodialysis, peritoneal dialysis and pre-emptive transplant. Dialysis and transplant era were classified in 10 year intervals.

## Statistical analysis

Baseline variables were summarised using counts and percentages and assessed by χ2 tests of independence with Bonferroni correction for multiple testing (S1 and S2 Tables). Continuous variables were assessed with one way Analysis of Variance (ANOVA) with Bonferroni correction. Results were considered statistically significant if p<0.005.

Median follow-up time for dialysis and transplant cohorts was calculated using reverse Kaplan-Meier estimator. Univariable and multivariable Cox proportional hazards models were used to calculate the association between exposures and covariables with outcome variables (mortality, kidney transplantation). In the dialysis cohort analyses, covariates included gender, ethnicity, smoking status, body mass index (BMI) diabetes status, first dialysis modality and dialysis vintage. In the transplant cohort analyses; recipient gender, ethnicity, age, smoking status BMI, comorbidities, first KRT modality, human leukocyte antigen (HLA) mismatch and transplant era were included as covariates. In the kidney transplantation analyses, recipient gender, ethnicity, age, smoking status BMI, comorbidities, first KRT modality, HLA mismatch and transplant era were included as covariates. Hazard ratios (HRs) and 95%

confident intervals (CIs) were calculated for each characteristic. Results were considered statistically significant if p<0.05.

In sensitivity analyses, Cox proportional hazard models were calculated for primary outcomes with non-ESKD were subclassified into diabetic nephropathy, glomerular disease, autosomal dominant polycystic kidney disease (ADPKD) and other kidney diseases (S1 Table). Diabetic nephropathy, glomerular disease and ADPKD were selected as each disease has known demographic features and outcomes. Hypertension was classified with other kidney diseases as it is unclear if hypertension is the cause of kidney failure or consequence an undiagnosed kidney disease [6]. Adjusted sub-distribution HRs (ASHRs) were generated using Fine and Gray's proportional hazards models where mortality and kidney transplantation were competing risks [7]. Results were considered statistically significant if p<0.05.

Only complete cases were included in the analyses. All analyses were conducted in SPSS software (IBM Corp. Released 2021. IBM SPSS Statistics for Windows, Version 28.0. Armonk, NY: IBM Corp).

## Results

### Participant demographics

Sixty thousand four hundred and forty-eight people on dialysis were included in the study with 3,640 people with uESKD, 24,513 people with diabetic nephropathy, 10,450 people with glomerular disease, 2,188 people with ADPKD and 19,657 with other kidney disease (Fig 1 and S2 Table). Twenty thousand eight hundred and fifty-nine people received kidney transplants–880 recipients had uESKD, 2,458 recipients had diabetic nephropathy, 8,957 recipients had glomerular disease, 2,831 recipients had ADPKD and 5,733 recipients had other kidney disease (S3 Table). Median follow-up times for the dialysis and transplant cohorts were 10 years and 14 years respectively.

### Mortality

In people on dialysis, 1 year, 3 year and 5 year mortality rates were 19.2%, 39.7% and 53.1% respectively (S2 Table). People on dialysis with uESKD had increased mortality risk compared to people with known-ESKD (AHR 1.10, 95% CI 1.06–1.16, Table 1) on multivariable analysis. On subgroup analysis, people with uESKD had similar mortality risk compared to other kidney diseases (S4 Table). Kidney transplant recipients had 1 year, 3 year and 5 year mortality rates of 0.6%, 2.4% and 4.7% respectively (S3 Table). Kidney transplant recipients with uESKD had increased mortality risk compared to those with known-ESKD (AHR 1.17, 95% CI 1.01–1.35, Table 2). On subgroup analysis, recipients with uESKD had similar mortality risk to those with other kidney diseases (S5 Table). Recipients with glomerular disease or ADPKD had reduced mortality risk while those with diabetic nephropathy had increased mortality risk.

### Kidney transplantation

People with uESKD on KRT had similar likelihood of kidney transplantation compared to people with known-ESKD (Table 3). On subgroup analysis, uESKD had similar kidney transplantation compared to people with other kidney diseases (S6 Table). Death censored kidney transplantation was increased in people with uESKD compared to people with known-ESKD (AHR 1.24, 95% CI 1.18–1.29, S7 Table). Demographic features such as age between 40–59 years old, BMI between 25–29.9, peritoneal dialysis as first KRT modality and more recent KRT initiation were associated with increased likelihood of kidney transplant. People of female

**Table 1. Unadjusted + adjusted hazard ratios + 95% CI for association between kidney disease and mortality in dialysis cohort.**

| Effect | Unadjusted | | Adjusted | |
|---|---|---|---|---|
| | HR | 95% CI | HR | 95% CI |
| **Disease status** | | | | |
| uESKD | 1.15*** | 1.11–1.20 | 1.10*** | 1.06–1.16 |
| Known-ESKD | Ref | | Ref | |
| **Gender** | | | | |
| Male | Ref | | Ref | |
| Female | 0.93*** | 0.91–0.95 | 1.01 | 1.05–1.1 |
| **Ethnicity** | | | | |
| White | Ref | | Ref | |
| Non-white | 0.75*** | 0.73–0.76 | 0.87*** | 0.85–0.89 |
| **Age** | | | | |
| < 20 Years | Ref | | Ref | |
| 20–39 Years | 0.72*** | 0.61–0.86 | 0.70*** | 0.58–0.84 |
| 40–59 Years | 1.30** | 1.09–1.54 | 1.18 | 0.98–1.4 |
| 60–79 Years | 1.84*** | 1.53–2.10 | 1.62*** | 1.35–1.94 |
| **Smoking status** | | | | |
| Never | Ref | | Ref | |
| Former | 1.17*** | 1.14–1.19 | 1.07** | 1.05–1.10 |
| Current | 1.07*** | 1.04–1.11 | 1.21*** | 1.17–1.25 |
| **BMI (kg/m2)** | | | | |
| <18.5 | Ref | | Ref | |
| 18.5–24.9 | 0.85*** | 0.81–0.90 | 0.76*** | 0.72–0.80 |
| 25–29.9 | 0.81*** | 0.77–0.85 | 0.71*** | 0.67–0.75 |
| >30 | 0.71*** | 0.67–0.75 | 0.67*** | 0.64–0.72 |
| **Comorbidities** | | | | |
| Diabetes mellitus | 1.03*** | 1.01–1.05 | 1.19*** | 1.16–1.21 |
| Coronary artery disease | 1.57*** | 1.54–1.60 | 1.27*** | 1.24–1.30 |
| Peripheral vascular disease | 1.49*** | 1.46–1.52 | 1.23*** | 1.20–1.26 |
| **First KRT modality** | | | | |
| Haemodialysis | Ref | | Ref | |
| Peritoneal dialysis | 1.03** | 1.01–1.05 | 1.01 | 0.99–1.04 |
| **Dialysis vintage** | | | | |
| 1989–1998 | Ref | | Ref | |
| 1999–2008 | 0.85*** | 0.83–0.87 | 0.85*** | 0.83–0.88 |
| 2009–2018 | 0.67*** | 0.66–0.69 | 0.69*** | 0.67–0.71 |
| 2018–2021 | 0.33*** | 0.31–0.36 | 0.37*** | 0.34–0.41 |

**Abbreviations**: BMI = body mass index, CI = confidence interval, HR = hazard ratio, KRT = kidney replacement therapy, ref = reference, uESKD = kidney failure of unknown aetiology.

Significance level:

*<0.05

**<0.01

***<0.001.

gender, age between 60–79 years old and current smoking status with comorbidities were associated with reduced likelihood of kidney transplant.

**Table 2. Unadjusted + adjusted hazard ratios and 95% CI for association between kidney disease status and mortality in transplant cohort.**

| Effect | Unadjusted | | Adjusted | |
|---|---|---|---|---|
| | HR | 95% CI | HR | 95% CI |
| **Recipient disease status** | | | | |
| uESKD | 1.17* | 1.03–1.33 | 1.17* | 1.01–1.35 |
| Known-ESKD | Ref | | Ref | |
| **Recipient gender** | | | | |
| Male | Ref | | Ref | |
| Female | 0.86*** | 0.82–0.91 | 0.96 | 0.90–1.02 |
| **Recipient ethnicity** | | | | |
| White | Ref | | Ref | |
| Non-white | 0.82*** | 0.77–0.87 | 0.81*** | 0.76–0.88 |
| **Recipient age** | | | | |
| < 20 Years | Ref | | Ref | |
| 20–39 Years | 1.56*** | 1.31–1.86 | 1.27*** | 1.03–1.58 |
| 40–59 Years | 4.11*** | 3.48–4.84 | 3.53*** | 2.87–4.34 |
| 60–79 Years | 8.1*** | 6.89–9.65 | 7.36*** | 5.95–9.10 |
| **Recipient smoking status** | | | | |
| Never | Ref | | Ref | |
| Former | 1.67*** | 1.57–1.77 | 1.22*** | 1.14–1.31 |
| Current | 1.75*** | 1.61–1.90 | 1.76*** | 1.62–0.92 |
| **Recipient BMI (kg/m$^2$)** | | | | |
| <18.5 | Ref | | Ref | |
| 18.5–24.9 | 1.58*** | 1.38–1.81 | 0.79** | 0.68–0.92 |
| 25–29.9 | 2.12*** | 1.85–2.44 | 0.84* | 0.72–0.99 |
| >30 | 2.40*** | 2.08–2.77 | 0.94 | 0.80–1.11 |
| **Recipient comorbidities** | | | | |
| Diabetes mellitus | 2.7*** | 2.61–2.98 | 2.03*** | 1.87–2.20 |
| Coronary artery disease | 2.58*** | 2.40–2.78 | 1.30*** | 1.19–1.42 |
| Peripheral vascular disease | 2.89*** | 2.63–3.17 | 1.48*** | 1.32–1.64 |
| **First KRT modality** | | | | |
| Haemodialysis | Ref | | Ref | |
| Peritoneal dialysis | 0.96 | 0.91–0.10 | 1.05 | 0.98–1.12 |
| Pre-emptive | 0.54*** | 0.47–0.61 | 0.73*** | 0.64–0.84 |
| **HLA mismatch** | | | | |
| 0 Mismatch | Ref | | Ref | |
| 1–3 Mismatch | 1.23*** | 1.08–1.40 | 1.25** | 1.08–1.44 |
| 4–6 Mismatch | 1.36*** | 1.19–1.55 | 1.35*** | 1.16–1.56 |
| **Transplant era** | | | | |
| 1989–1998 | Ref | | Ref | |
| 1999–2008 | 0.74*** | 0.70–0.79 | 0.68*** | 0.63–0.73 |
| 2009–2018 | 0.65*** | 0.60–0.711 | 0.42*** | 0.38–0.46 |
| 2018–2021 | 0.38*** | 0.27–0.52 | 0.21*** | 0.15–0.29 |

**Abbreviations**: BMI = body mass index, HLA = human leukocyte antigen, KRT = kidney replacement therapy, ref = reference, uESKD = kidney failure of unknown aetiology.

Significance level:

*<0.05

**<0.01

***<0.001.

**Table 3. Unadjusted + adjusted hazard ratios + 95% CI for association between kidney disease status and kidney transplantation.**

| Effect | Unadjusted | | Adjusted | |
|---|---|---|---|---|
| | HR | 95% CI | HR | 95% CI |
| **Disease status** | | | | |
| uESKD | 1.01 | 0.94–1.08 | 0.99 | 0.94–1.08 |
| Known-ESKD | Ref | | Ref | |
| **Gender** | | | | |
| Male | Ref | | Ref | |
| Female | 0.85*** | 0.83–0.88 | 0.84*** | 0.81–0.86 |
| **Ethnicity** | | | | |
| White | Ref | | Ref | |
| Non-white | 1.00 | 0.97–1.03 | 0.92*** | 0.89–0.95 |
| **Age** | | | | |
| < 20 Years | Ref | | Ref | |
| 20–39 Years | 0.87*** | 0.82–0.92 | 0.97 | 0.95–1.1 |
| 40–59 Years | 1.26*** | 1.19–1.32 | 1.38*** | 1.30–1.46 |
| 60–79 Years | 0.89*** | 0.844–0.95 | 0.86** | 0.85–0.97 |
| **Smoking status** | | | | |
| Never | Ref | | Ref | |
| Former | 1.03 | 0.99–1.06 | 1.02 | 0.99–1.06 |
| Current | 0.83*** | 0.80–0.88 | 0.89*** | 0.85–0.94 |
| **BMI (kg/m2)** | | | | |
| <18.5 | Ref | | Ref | |
| 18.5–24.9 | 1.04 | 0.98–1.1 | 1.05 | 0.98–1.1 |
| 25–29.9 | 1.27*** | 1.19–1.35 | 1.14*** | 1.06–1.22 |
| >30 | 1.28*** | 1.2–1.36 | 1.05 | 0.97–1.1 |
| **Comorbidities** | | | | |
| Diabetes mellitus | 0.76*** | 0.73–0.79 | 0.62*** | 0.60–0.65 |
| Coronary artery disease | 0.68*** | 0.65–0.71 | 0.80*** | 0.76–0.84 |
| Peripheral vascular disease | 0.62*** | 0.58–0.66 | 0.83*** | 0.78–0.89 |
| **First KRT modality** | | | | |
| Haemodialysis | Ref | | Ref | |
| Peritoneal dialysis | 1.2*** | 1.20–1.28 | 1.24*** | 1.20–1.28 |
| Pre-emptive | 2.1*** | 2.0–2.2 | 1.54*** | 1.46–0.1.61 |
| **KRT onset year** | | | | |
| 1989–1998 | Ref | | Ref | |
| 1999–2008 | 3.07*** | 2.9–3.2 | 3.6*** | 3.4–3.8 |
| 2009–2018 | 23.8*** | 22.4–25.41 | 32.7*** | 30.5–35.0 |
| 2018–2021 | 304.5*** | 266.7–347.7 | 468.8*** | 406.4–540.9 |

**Abbreviations**: ATSI = Aboriginal and Torres Strait Islander, BMI = body mass index, HR = hazard ratio, KRT = kidney replacement, therapy, ref = reference, uESKD = kidney failure of unknown aetiology.

Significance level:

*<0.05

**<0.01

***<0.001.

## Discussion

This study showed that people with uESKD have increased mortality risk but similar likelihood of kidney transplantation compared to people with known-ESKD. The prevalence of uESKD in people on dialysis and transplant was 6.0% and 4.2% respectively, which is lower than rates in United Kingdom (14.9%) [8], Europe (17.0%) [9], Brazil (24%) [10] and Mexico (54%) [3]. This difference in uESKD prevalence is likely multifactorial in the context of different occupational and environmental exposures; and access/utilisation of advanced diagnostic tests such as genetic testing. The high prevalence of uESKD in Mexico has been linked to intense work in strong heat, increased environmental degradation with exposure to heavy metals, widespread use of pesticides and reduced access to diagnostic testing to identify the cause of kidney failure [11].

People with uESKD on dialysis had increased mortality risk compared to people with known-ESKD. On subgroup analysis, uESKD had increased mortality risk compared to diabetic nephropathy, glomerular disease and ADPKD. Reasons for this finding is likely multifactorial–absence of cause-specific treatment for extra-kidney manifestations, older age at KRT initiation and socioeconomic factors. These results are different to those reported by Gutierrez-Peña *et al*. where people in Aguascalientes, Mexico, on KRT with uESKD had superior survival compared to those with known-ESKD on age-adjusted analyses [3]. In the aforementioned study, a significant proportion of known-ESKD participants had diabetic nephropathy which was associated with inferior mortality outcomes compared to people with other causes of kidney failure [3].

Kidney transplant recipients with uESKD have increased mortality risk compared to those known-ESKD. On subgroup analysis, uESKD performed similarly compared to other kidney diseases. Glomerular disease and ADPKD were associated with superior post-transplant mortality outcomes compared to those with other kidney diseases–likely contributing to the outcomes seen in the binary (uESKD vs. known-ESKD) exposure analyses. A previous ANZDATA analysis identified that transplant recipients with uESKD had similar mortality risk compared to recipients with commonly-recurring glomerular diseases [12]. Commonly-recurring glomerular diseases carry higher mortality risks associated with increased risk of graft failure and higher immunosuppression burden [12]. Graft failure data was not accessible to test this hypothesis. The finding that recipients with uESKD have similar mortality risk compared to other kidney diseases was also identified in an USRDS study of younger transplant recipients [13].

People on KRT with uESKD had similar likelihood of kidney transplantation compared to people with known-ESKD. On death-censored kidney transplantation, people with uESKD had higher kidney transplantation rates compared to people with known-ESKD suggesting that the increased mortality risk of people with uESKD may be contributing to the results seen in the headline analyses. All people on KRT were included in the kidney transplantation analyses, however an unknown proportion would have been deemed unsuitable for transplantation. As such, it was not possible to assess kidney transplantation solely in those who were suitable for transplantation. Subset analysis of patients suitable for transplantation will be possible in the future with the recent addition of "suitability for kidney transplant" in ANZDATA data collection.

In this study, uESKD performed similarly to other causes of kidney failure in subgroup analyses for demographics, mortality and kidney transplantation, suggesting that uESKD may overlap with conditions in the "other kidney disease" category. Chronic kidney disease of uncertain aetiology (CKDu) observed in low and middle income countries mainly occurs in agricultural communities affecting young males [14]. In our analyses, uESKD was associated with increased age which may be due to reduced appetite for higher risk diagnostic procedures

such as kidney biopsies in older people with atrophic kidneys [15]. This disparity further signals that uESKD as recorded in ANZDATA is different to CKDu reported elsewhere and that uESKD is highly jurisdiction-dependent. Further study is required to elucidate the potential genetic, occupational, and environmental factors causing uESKD in Australia and New Zealand.

Limitations included the use of retrospective observational data, thereby confounded by measurement bias, and unmeasured factors not collected by ANZDATA. Primary kidney disease classifications in ANZDATA are based on clinician classification as the dominant cause, and are not always biopsy- or genetically-proven. Furthermore, advances in diagnostic approaches and disparities in access to such diagnostics can lead to inconsistencies in uESKD definition across regions and over time.

## Conclusions

People with uESKD on KRT had increased mortality risks compared to known-ESKD. uESKD has similar likelihood of kidney transplantation compared to known-ESKD. On subgroup analysis, the uESKD group had similar demographic features compared to other kidney diseases and performed similarly on outcome measures, suggesting that uESKD may include people with "other kidney diseases". Further studies are required to confirm this hypothesis and correlated uESKD recorded in ANZDATA to CKDu in other jurisdictions.

## Supporting information

**S1 Table. Breakdown of kidney failure causes in other kidney disease group.**
(DOCX)

**S2 Table. Characteristics and medical conditions of the dialysis cohort.**
(DOCX)

**S3 Table. Characteristics and medical conditions of the transplant cohort.**
(DOCX)

**S4 Table. Subgroup analysis evaluating association between kidney disease status and mortality in dialysis cohort.**
(DOCX)

**S5 Table. Subgroup analysis evaluating association between kidney disease status and mortality in transplant cohort.**
(DOCX)

**S6 Table. Subgroup analysis evaluating association between kidney disease status and kidney transplantation in KRT cohort.**
(DOCX)

**S7 Table. Death-censored kidney transplantation competing risk analysis.**
(DOCX)

**S8 Table. Modified STROBE statement.**
(DOCX)

## Acknowledgments

The authors are grateful for the significant contributions of the Australian and New Zealand nephrology community (physicians, surgeons, database managers, nurses, people receiving

KRT) in providing information for and maintaining the ANZDATA database. The data that supports the findings of this study were obtained from the ANZDATA–data request ID 42579. Restrictions apply to the availability of these data, which were used under license for this study. Data are available from www.anzdata.org.au, subject to the registry's data release policies. MSYN acknowledges the Robert and Janelle Bird Postdoctoral Research Fellowship. AJM acknowledges support from a Queensland Health Advancing Clinical Research Fellowship.

## Author Contributions

**Conceptualization:** Lucy S. Wang, Andrew J. Mallett.

**Data curation:** Lucy S. Wang, Monica S. Y. Ng, Andrew J. Mallett.

**Formal analysis:** Lucy S. Wang, Venkat Vangaveti, Monica S. Y. Ng, Andrew J. Mallett.

**Investigation:** Lucy S. Wang, Andrew J. Mallett.

**Methodology:** Lucy S. Wang, Venkat Vangaveti, Monica S. Y. Ng, Andrew J. Mallett.

**Project administration:** Monica S. Y. Ng, Andrew J. Mallett.

**Resources:** Andrew J. Mallett.

**Supervision:** Monica S. Y. Ng, Andrew J. Mallett.

**Validation:** Venkat Vangaveti.

**Visualization:** Lucy S. Wang, Venkat Vangaveti.

**Writing – original draft:** Lucy S. Wang, Venkat Vangaveti, Monica S. Y. Ng, Andrew J. Mallett.

**Writing – review & editing:** Lucy S. Wang, Venkat Vangaveti, Monica S. Y. Ng, Andrew J. Mallett.

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
