## [Decision Letter · Decision Letter 0]

26 Nov 2023

PONE-D-23-34616Characteristics and clinical outcomes of patients with kidney failure of unknown aetiology from ANZDATA registryPLOS ONE

Dear Dr. Mallett,

Thank you for submitting your manuscript to PLOS ONE. After careful consideration, we feel that it has merit but does not fully meet PLOS ONE’s publication criteria as it currently stands. Therefore, we invite you to submit a revised version of the manuscript that addresses the points raised during the review process.

We look forward to receiving your revised manuscript.

Kind regards,

Mohamed E Elrggal

Academic Editor

PLOS ONE

Journal Requirements:

   "I have read the journal's policy and the authors of this manuscript have the following competing interests: "M.S.Y.N. has received research grants and travel sponsorships from Avant Foundation and postdoctoral research fellowship from Royal Brisbane and Women’s Hospital Foundation."

Additional Editor Comments:

Thank you for submitting your manuscript to PLOS ONE, after careful reviewing, our peer reviewers decided that your manuscript needs further revision.  

Reviewers' comments:

Reviewer's Responses to Questions

**Comments to the Author**

1. Is the manuscript technically sound, and do the data support the conclusions?

Reviewer #1: Yes

Reviewer #2: Partly

2. Has the statistical analysis been performed appropriately and rigorously? 

Reviewer #1: Yes

Reviewer #2: No

3. Have the authors made all data underlying the findings in their manuscript fully available?

Reviewer #1: Yes

Reviewer #2: Yes

4. Is the manuscript presented in an intelligible fashion and written in standard English?

Reviewer #1: Yes

Reviewer #2: Yes

5. Review Comments to the Author

Reviewer #1: Reviewers comments

Characteristics and clinical outcomes of patients with kidney failure of unknown etiology from ANZDATA registry

General comment

1. Well done to the authors for submitting this paper for possible publication

2. It will helpful to come up clearly with the definition of uESKD. It should have a diagnostic criteria clearly similar to literature and other studies authors hope to compare their study to.

3. Most sentences need a senior author to read through to conform to manuscript standards

4. CKDu is a fairly new area or study but there has been a lot of work with clear definitions I was hoping authors will explore

5. Sentences should not be started with roman numerals

Specific comments

Abstract

1. What is eESKD exactly. In terms of definition?

2. Law of first mention for uESKD and how was it defined?

3. ‘Dependent on the region, 16% of chronic kidney disease’…. Authors should state exactly the region they quoted.

4. Smoking cannot be considered as a cause of CKD and hence should stay as a risk factor.

Material and methods

5. Was wondering why the data set ends in 2021. Could the authors add on up to at least 2022?

6. It might be helpful to define clearly all terms used in the study here.

Results

7. I suggest autors do not start sentences with numbers in manuscript

8. “Age 40-59 years old, BMI 25-29.9, peritoneal dialysis as first KRT

modality and more recent KRT..”Most sentences can be well written as academic writing to make it easier to read. Like the example above. Many such in the write up

9. ‘…current smoking status and presence of comorbidities were associated with reduced likelihood of kidney transplant. Did authors assess the likelihood ratio?

Discussion

10. which is lower than rates in United Kingdom (14.9%) [7], Europe (17.0%) [8], Brazil (24%) [9] and Mexico (54%) [10]. It will be helpful to discuss why Mexico had such high prevalence or what makes the studies different from the UK, Europe and Brazil.

11. “People with uESKD on dialysis had increased mortality risk compared to people with non-uESKD. On subgroup analysis, uESKD had increased mortality risk compared to diabetic nephropathy, glomerular disease and ADPKD; but similar mortality risk compared to other kidney diseases.” It might be helpful to avoid long sentences of the results in the discussion.

12. Chronic kidney disease of uncertain aetiology (CKDu) observed in low and middle income countries mainly occurs in agricultural communities affecting young males [13]. It is fundamentally helpful to know the clear definition of uESKD in the registry. Did it meet a clear diagnostic criteria or just physician clinical judgement? Very difficult to compare to others with clear definition. …and are not always biopsy- or genetically-proven.” What proportions were biopsy proven then?” Will the findings be different with biopsy proven diagnosis?

13. Further study is required to elucidate the potential genetic, occupational, and environmental factors causing uESKD in Australia. Are there no reports at all or studies in New Zealand or Australia on the subject?

14. “Shortcomings included the use of retrospective observational data…”. I suggest limitation to the study might be a preferred term.

Conclusion

15. ‘People with uESKD on KRT had increased mortality risks compared to non-uESKD. uESKD has similar progression to kidney transplantation compared to uESKD’. Not clear in my mind the clear importance of this descriptive studies? what hypothesis or research question are authors seeking to generate or hoping to answer?

Reviewer #2: The manuscript entitled (Characteristics and clinical outcomes of patients with kidney failure of unknown aetiology from ANZDATA registry) discusses a quite important issue and highlights the prevalence and outcomes of ESRD of unknown etiology and these are my comments:

1- Abstract: correct the typo in (eESKD) in the 4th line of the introduction.

2- Materials and Methods, Statistical analysis:

a) You mentioned that baseline variables were summarised using counts and percentages and assessed by χ2 tests of independence (Table 1 and 2). In these tables, there are a lot of significant associations, so you need to perform the Bonferroni correction for a chi-square analysis for multiple comparisons.

b) In Table S2, there are some continuous variables (for example; dialysis vintage) you should mention the statistical test used.

3- Please mention what was the basis of the classification of causes of ESKD and why hypertension was not considered in your classifications.

4- Other causes of ESKD constituted a large number of your cohort. They were approximately one-third of the dialysis cohort. Please identify these causes and the percentage of each cause.

5- do you have actual numbers for cases with definite diagnosis by renal biopsy or genetic testing?

6- In my opinion, the term ''progression'' to kidney transplant is not proper and it may be better to change it to receiving kidney transplant.

6. PLOS authors have the option to publish the peer review history of their article (what does this mean?). If published, this will include your full peer review and any attached files.

Reviewer #1: No

Reviewer #2: No

---

## [Author Response · Author response to Decision Letter 0]

30 Dec 2023

Responses to Editorial and Reviewer Queries PONE-D-23-34616 “Characteristics and clinical outcomes of patients with kidney failure of unknown aetiology from ANZDATA registry”

Editorial Team

 - We have reviewed and completed this. 

 "I have read the journal's policy and the authors of this manuscript have the following competing interests: "M.S.Y.N. has received research grants and travel sponsorships from Avant Foundation and postdoctoral research fellowship from Royal Brisbane and Women’s Hospital Foundation."

 - This has now been updated to the following and we are thankful for your actioning in the online submission form:

“Conflict of Interest Statement

M.S.Y.N. has received research grants and travel sponsorships from Avant Foundation and postdoctoral research fellowship from Royal Brisbane and Women’s Hospital Foundation. This does not alter our adherence to PLOS ONE policies on sharing data and materials.”

 - Similar to other analyses published in PLOS ONE using data provided by the ANZDATA registry, the Data Availability statement has been updated to:

“Data Availability Statement

The authors confirm that all data underlying the findings are fully available upon request and without restriction. The primary dataset for this manuscript was generated and made available to the authors by the Australia and New Zealand Dialysis and Transplant (ANZDATA) Registry, Adelaide, Australia. Data used in this study belongs to the ANZDATA registry. Data stored in ANZDATA is collected and stored on behalf of all Australian and New Zealand renal units. The ANZDATA data usage agreement between the ANZDATA Registry and the authors does not allow the authors to make the data publicly available. The authors confirm that all data underlying the findings can be obtained without restriction directly from the ANZDATA Registry on request (email address requests@anzdata.org.au, website https://www.anzdata.org.au/anzdata/services/data-policies/). The authors of this paper did not access the data via special access privileges and only gained access to the data after the data request was approved by the ANZDATA.”

This Data Availability statement with PLOS data policies and aligns to previous recently published PLOS ONE articles utilising the same data source under the same circumstances and ANZDATA Data Access Agreement:

• https://doi.org/10.1371/journal.pone.0236396

• https://doi.org/10.1371/journal.pone.0293721

• https://doi.org/10.1371/journal.pone.0249000

 - This has been actioned. 

 

Reviewer #1

General comment

Well done to the authors for submitting this paper for possible publication

1. It will helpful to come up clearly with the definition of uESKD. It should have a diagnostic criteria clearly similar to literature and other studies authors hope to compare their study to.

 - Thank you for your comment. Notably, kidney failure of unknown aetiology (uESKD) in ANZDATA is denoted by treating kidney specialist. By definition, uESKD in ANZDATA is where there is no known cause for kidney failure which involves exclusion of other causes of kidney failure such as hypertension, diabetic nephropathy, glomerular disease and genitourinary disease. This definition is similar to the definition used by Gutierrez-Peña et al. in their study of uESKD in Mexico (CKJ 2021, 14(4): 1197-1206). The definition of uESKD has been clarified in the manuscript: Primary exposure was kidney disease type classified as kidney failure of unknown aetiology (uESKD) or non-uESKD based on kidney failure cause codes in ANZDATA. In ANZDATA, cause of kidney failure is denoted by treating kidney specialist based on clinical features and may not be biopsy proven. Cases not caused by diabetic nephropathy, glomerular disease, hypertension or any other identifiable cause are classified as uESKD in ANZDATA. 

2. Most sentences need a senior author to read through to conform to manuscript standards

 - Sentences in the manuscript has been adjusted as suggested. 

3. CKDu is a fairly new area or study but there has been a lot of work with clear definitions I was hoping authors will explore.

 - CKDu is a diagnosis of exclusion – where there is no other cause of kidney disease identified. While current articles propose potential causes of CKDu such as nephrotoxins (e.g. lead, cadmium, aristolochic acid, arsenic, other environmental causes) – there is no clear cause and therefore no clear definition (BMC nephrology 2015, 16:145). In population studies, CKDu is characterised by kidney failure in young and middle aged adults with male predominance in low income agricultural communities with absence of known causes of chronic kidney disease. People with CKDu have bland urine sediment, minimal proteinuria and interstitial fibrosis +/- non-specific glomerular damage on biopsy (BMC nephrology 2015, 16:145; CKJ 2021, 14(4): 1197-1206). 

Notably, Australia/New Zealand is geopolitically and socioeconomically quite different to locales with CKDu and one of the key findings of this study is that uESKD as defined in ANZDATA is not the same as CKDu seen low income agricultural communities. It may include some CKDu by definition – “kidney disease of known aetiology” but the demographic and outcome data suggest that uESKD is more likely to align the other kidney diseases group in ANZDATA. This is directly stated in our discussion:

In this study, uESKD performed similarly to other causes of kidney failure in subgroup analyses for demographics, mortality and progression to transplant, suggesting that uESKD may overlap with conditions in the “other kidney disease” category. Chronic kidney disease of uncertain aetiology (CKDu) observed in low and middle income countries mainly occurs in agricultural communities affecting young males [13]. In our analyses, uESKD was associated with increased age which may be due to reduced appetite for higher risk diagnostic procedures such as kidney biopsies in older people with atrophic kidneys [14]. This disparity further signals that uESKD as recorded in ANZDATA is different to CKDu reported elsewhere and that uESKD is highly jurisdiction-dependent. Further study is required to elucidate the potential genetic, occupational, and environmental factors causing uESKD in Australia.

4. Sentences should not be started with roman numerals

 - Sentence beginning with roman numerals have been amended. 

Specific comments

Abstract

1. What is uESKD exactly. In terms of definition?

 - Kidney failure of unknown aetiology (uESKD) in ANZDATA is defined as causes of kidney failure where there is no cause found which involves exclusion of other causes of kidney failure such as hypertension, diabetic nephropathy, glomerular disease and genitourinary disease. This diagnosis is denoted by treating kidney specialist. This definition is similar to the definition used by Gutierrez-Peña et al. in their study of uESKD in Mexico (CKJ 2021, 14(4): 1197-1206). The definition of uESKD has been clarified in the manuscript: Primary exposure was kidney disease type classified as kidney failure of unknown aetiology (uESKD) or non-uESKD based on kidney failure cause codes in ANZDATA. Cases not caused by diabetic nephropathy, glomerular disease, hypertension or any other identifiable cause are classified as uESKD in ANZDATA. In ANZDATA, cause of kidney failure is denoted by treating kidney specialist based on clinical features and may not be biopsy- nor genetically- proven.

2. Law of first mention for uESKD and how was it defined?

 - uESKD is now defined in the second sentence as suggested: uESKD is defined as kidney failure cases where traditional risk factors, including diabetes, hypertension, smoking and obesity, and other primary renal diseases are excluded as potential causes

3. ‘Dependent on the region, 16% of chronic kidney disease’…. Authors should state exactly the region they quoted.

 - In the original sentence the statistic was from India (BMC Nephrology 2015, 16:145). This sentence has been adjusted to the following to report on uESKD to better match with content of article: Kidney failure of unknown aetiology (uESKD) is also heavily location dependent varying between 27% in Egypt to 54% in Aguacalientes, Mexico [1-3].

4. Smoking cannot be considered as a cause of CKD and hence should stay as a risk factor.

 - This sentence has been altered to the following as suggested: [1-3]. uESKD is defined as kidney failure cases where there other causes of kidney diseases such as diabetes, hypertension, glomerular disease have been excluded as potential causes.

Material and methods

1. Was wondering why the data set ends in 2021. Could the authors add on up to at least 2022?

 - The data set ends at 2021 as ANZDATA is requires approximately 2 years for data to be available. Unfortunately, it is not possible to add data from 2022 as that data would not be available until next year. 

2. It might be helpful to define clearly all terms used in the study here.

 - Terms have been clarified in the variables section of the article. Notably, kidney disease and comorbidity are denoted by treating kidney specialist. 

Results

1. I suggest autors do not start sentences with numbers in manuscript

 - This has been corrected as requested in the article. 

2. “Age 40-59 years old, BMI 25-29.9, peritoneal dialysis as first KRT modality and more recent KRT..”Most sentences can be well written as academic writing to make it easier to read. Like the example above. Many such in the write up

 - The above sentences have been corrected to: Demographic features such as age between 40-59 years old, BMI between 25-29.9, peritoneal dialysis as first KRT modality and more recent KRT initiation were associated with increased likelihood of kidney transplant. People of female gender, age between 60-79 years old and current smoking status with comorbidities were associated with reduced likelihood of kidney transplant.

3. ‘…current smoking status and presence of comorbidities were associated with reduced likelihood of kidney transplant. Did authors assess the likelihood ratio?

 - We assessed hazard ratio for association of demographic factors with outcomes of mortality and kidney transplant as we used Cox Proportional Hazard Regression modelling. These are reported in tables 1-3.

Discussion

1. Which is lower than rates in United Kingdom (14.9%) [7], Europe (17.0%) [8], Brazil (24%) [9] and Mexico (54%) [10]. It will be helpful to discuss why Mexico had such high prevalence or what makes the studies different from the UK, Europe and Brazil.

 - The following sentence has been added to explain the high prevalence of uESKD in Mexico: The high prevalence of uESKD in Mexico has been linked to intense work in strong heat, increased environmental degradation with exposure to heavy metals, widespread use of pesticides and reduced access to diagnostic testing to identify the cause of kidney failure [10].

2. “People with uESKD on dialysis had increased mortality risk compared to people with non-uESKD. On subgroup analysis, uESKD had increased mortality risk compared to diabetic nephropathy, glomerular disease and ADPKD; but similar mortality risk compared to other kidney diseases.” It might be helpful to avoid long sentences of the results in the discussion.

 - The sentences have been shortened as requested: People with uESKD on dialysis had increased mortality risk compared to people with non-uESKD. On subgroup analysis, uESKD had increased mortality risk compared to diabetic nephropathy, glomerular disease and ADPKD.

3. Chronic kidney disease of uncertain aetiology (CKDu) observed in low and middle income countries mainly occurs in agricultural communities affecting young males [13]. It is fundamentally helpful to know the clear definition of uESKD in the registry. Did it meet a clear diagnostic criteria or just physician clinical judgement? Very difficult to compare to others with clear definition. …and are not always biopsy- or genetically-proven.” What proportions were biopsy proven then?” Will the findings be different with biopsy proven diagnosis?

 - As requested, we have clarified the definition of uESKD in ANZDATA in the methods section: Cases not caused by diabetic nephropathy, glomerular disease, hypertension or any other identifiable cause are classified as uESKD in ANZDATA. In ANZDATA, cause of kidney failure is denoted by treating kidney specialist based on clinical features and may not be biopsy- nor genetically- proven. The purpose of this study was to describe uESKD as reported in ANZDATA registry. We have noted in the discussion that uESKD as described in the ANZDATA registry is different to CKDu observed in low and middle income countries as the demographic is more consistent with unclassified “other” kidney diseases. 

The proportion of diagnoses in this analysis that are biopsy proven are 23,640 (27.6%). We have reported number of biopsy proven diagnoses in each category in Supplementary Tables S1 and S2. Notably, many diagnoses (e.g. ADPKD, CAKUT, reflux nephropathy, diabetic nephropathy) are not diagnosed by biopsy – completing subgroup analyses on the biopsy-proven cohort would result in significant selection bias. 

4. Further study is required to elucidate the potential genetic, occupational, and environmental factors causing uESKD in Australia. Are there no reports at all or studies in New Zealand or Australia on the subject?

 - There are no linkage studies investigating genetic, occupational, environmental factors associated uESKD as described in ANZDATA which is different to CKDu as discussed previously.

5. “Shortcomings included the use of retrospective observational data…”. I suggest limitation to the study might be a preferred term.

 - Shortcomings has been changed to limitations as requested. 

Conclusion

1. ‘People with uESKD on KRT had increased mortality risks compared to non-uESKD. uESKD has similar progression to kidney transplantation compared to uESKD’. Not clear in my mind the clear importance of this descriptive studies? what hypothesis or research question are authors seeking to generate or hoping to answer?

 - There is limited information on clinical outcomes of people with uESKD in Australia and New Zealand – particularly at a population level. There was concern considering that as the cause of uESKD is unknown; recurrence risk after kidney transplant is also unknown – potentially leading to reluctance for physicians to list patients for kidney transplant. Reassuringly, people with uESKD have similar transplantations rates as people with ESKD with a known cause. Furthermore, information regarding mortality risk in people with uESKD is important for counselling discussions. This is discussed in the second paragraph of the discussion. We have also added a sentence to discuss our hypotheses: We hypothesised that people with uESKD would have similar mortality risk and reduced kidney transplantation rates compared to people with non-uESKD.

 

Reviewer #2

The manuscript entitled (Characteristics and clinical outcomes of patients with kidney failure of unknown aetiology from ANZDATA registry) discusses a quite important issue and highlights the prevalence and outcomes of ESRD of unknown etiology and these are my comments:

1. Abstract: correct the typo in (eESKD) in the 4th line of the introduction.

 - This this been corrected as suggested.

2. Materials and Methods, Statistical analysis:

a) You mentioned that baseline variables were summarised using counts and percentages and assessed by χ2 tests of independence (Table S1 and S2). In these tables, there are a lot of significant associations, so you need to perform the Bonferroni correction for a chi-square analysis for multiple comparisons.

 - We performed Bonferroni correction as requested. This has been added to the methods section. 

b) In Table S2, there are some continuous variables (for example; dialysis vintage) you should mention the statistical test used.

 - Continuous variables were assessed using χ2 tests of independence reported with Bonferroni correction for multiple testing.

3. Please mention what was the basis of the classification of causes of ESKD and why hypertension was not considered in your classifications.

 - The basis of the subgroup classification was to divide non-uESKD into subgroups with known demographic features and outcomes such as diabetic nephropathy, glomerular disease and ADPKD. In contrast, hypertension is a non-specific subgroup where it is unclear if hypertension is the cause or consequence of disease processes. This has been included in the methods section: Diabetic nephropathy, glomerular disease and ADPKD were selected as each disease has known demographic features and outcomes. Hypertension was classified with other kidney diseases as it is unclear if hypertension is the cause of kidney failure or consequence an undiagnosed kidney disease [6].

4. Other causes of ESKD constituted a large number of your cohort. They were approximately one-third of the dialysis cohort. Please identify these causes and the percentage of each cause.

 - We have included causes of other ESKD in Table S1. 

5. Do you have actual numbers for cases with definite diagnosis by renal biopsy or genetic testing?

 - We have included number of biopsy-proven cases in Table S2 and S3. We do not have data on genetically-proven cases. 

6. In my opinion, the term ''progression'' to kidney transplant is not proper and it may be better to change it to receiving kidney transplant.

 - Thank you for your feedback. We have removed the word “progression” from kidney transplantation terminology in text.

---

## [Decision Letter · Decision Letter 1]

19 Feb 2024

PONE-D-23-34616R1Characteristics and clinical outcomes of patients with kidney failure of unknown aetiology from ANZDATA registryPLOS ONE

Dear Dr. Mallett,

Thank you for submitting your manuscript to PLOS ONE. After careful consideration, we feel that it has merit but does not fully meet PLOS ONE’s publication criteria as it currently stands. Therefore, we invite you to submit a revised version of the manuscript that addresses the points raised during the review process.

**Thank you for response to our reviewers' comments. Still, some minor adjustments need to be made before final acceptance. Please address the reviewers' comments and send the manuscript back. **

We look forward to receiving your revised manuscript.

Kind regards,

Mohamed E Elrggal

Academic Editor

PLOS ONE

Journal Requirements:

Reviewers' comments:

Reviewer's Responses to Questions

**Comments to the Author**

1. If the authors have adequately addressed your comments raised in a previous round of review and you feel that this manuscript is now acceptable for publication, you may indicate that here to bypass the “Comments to the Author” section, enter your conflict of interest statement in the “Confidential to Editor” section, and submit your "Accept" recommendation.

Reviewer #1: All comments have been addressed

Reviewer #2: All comments have been addressed

2. Is the manuscript technically sound, and do the data support the conclusions?

Reviewer #1: Yes

Reviewer #2: Yes

3. Has the statistical analysis been performed appropriately and rigorously? 

Reviewer #1: Yes

Reviewer #2: Yes

4. Have the authors made all data underlying the findings in their manuscript fully available?

Reviewer #1: Yes

Reviewer #2: Yes

5. Is the manuscript presented in an intelligible fashion and written in standard English?

Reviewer #1: Yes

Reviewer #2: Yes

6. Review Comments to the Author

Reviewer #1: Thank you for the responses and congratulations to the authors.

Most comments have been well addressed.

I have no further comments for the authors

Reviewer #2: Thanks to the authors for the corrections and modifications that were made. I have few additional comments.

1- You classified them as uESKD and non-uESKD. In my opinion, it would sound better to replace non-uESKD with known ESKD.

2- In the results, you stated that the median follow-up time for the dialysis was 3 years. This indicates that at least 50% of the patients were followed for only 3 years. Then you mentioned that the mortality rate for people on dialysis at 5 years was 53.1%. I think these numbers need to be revised.

3- In conclusion; uESKD has similar likelihood of kidney transplantation compared to uESKD. Please revise and correct this sentence.

7. PLOS authors have the option to publish the peer review history of their article (what does this mean?). If published, this will include your full peer review and any attached files.

Reviewer #1: **Yes: **Dr. Elliot Koranteng Tannor

Reviewer #2: No

---

## [Author Response · Author response to Decision Letter 1]

22 Feb 2024

Responses to Editorial and Reviewer Queries PONE-D-23-34616R1 

“Characteristics and clinical outcomes of patients with kidney failure of unknown aetiology from ANZDATA registry”

Editorial Team

1. Thank you for response to our reviewers' comments. Still, some minor adjustments need to be made before final acceptance. Please address the reviewers' comments and send the manuscript back. 

Thank you so much for your and the Reviewers’ time and input which we greatly value. We have addressed the additional queries below.

Reviewer #1

2. Thank you for the responses and congratulations to the authors. Most comments have been well addressed. I have no further comments for the authors.

We appreciate your input and previous queries.

Reviewer #2

3. You classified them as uESKD and non-uESKD. In my opinion, it would sound better to replace non-uESKD with known ESKD.

This is a good point which we take on board. The first instance of mentioning this in the abstract and main manuscript we have used the term “non-uESKD (known-ESKD)“ to signify what this group is and then used “known-ESKD” thereafter. 

4. In the results, you stated that the median follow-up time for the dialysis was 3 years. This indicates that at least 50% of the patients were followed for only 3 years. Then you mentioned that the mortality rate for people on dialysis at 5 years was 53.1%. I think these numbers need to be revised.

Thank you for astutely pointing this out. We have investigated further and recalculated the median follow-up times using the reverse Kaplan-Meier estimator and have updates the results and statistical methodology accordingly. This methodology is more appropriate for calculating median follow-up times in cohorts such as this, and we thank the reviewer for identifying this. (New methods for estimating follow-up rates in cohort studies - PMC (nih.gov, PharmaSUG-2019-ST-081.pdf)

5. In conclusion; uESKD has similar likelihood of kidney transplantation compared to uESKD. Please revise and correct this sentence.

We have corrected this.

---

## [Editor Report · Decision Letter 2]

26 Feb 2024

Characteristics and clinical outcomes of patients with kidney failure of unknown aetiology from ANZDATA registry

PONE-D-23-34616R2

Dear Dr. Mallett,

We’re pleased to inform you that your manuscript has been judged scientifically suitable for publication and will be formally accepted for publication once it meets all outstanding technical requirements.

Kind regards,

Mohamed E Elrggal

Academic Editor

PLOS ONE
---

## [Editor Report · Acceptance letter]

1 Mar 2024

PONE-D-23-34616R2 

PLOS ONE

Dear Dr. Mallett, 

I'm pleased to inform you that your manuscript has been deemed suitable for publication in PLOS ONE. Congratulations! Your manuscript is now being handed over to our production team.

Kind regards, 

on behalf of

Dr. Mohamed E Elrggal 

Academic Editor

PLOS ONE